# Key Concepts in the Poliheuristic Theory of Foreign Policy Decision Making: A Comparative Examination Using Systemist Theory

**Steven B. Redd**

Department of Political Science, University of Wisconsin-Milwaukee, Milwaukee, WI 53201, USA;
sredd@uwm.edu

**Abstract:** The poliheuristic theory of foreign policy decision making incorporates the conditions surrounding foreign policy decisions, as well as the cognitive processes decision makers undergo en route to a choice. It argues that high-level decision makers, who routinely face stressful decision environments, engage in a two-stage decision process wherein they first employ cognitive-based, heuristic shortcuts in an attempt to simplify the decision task. In the second stage, once the decision task is more manageable, decision makers employ more analytic strategies in order to minimize risks and maximize rewards. Poliheuristic theory also posits that politics is the essence of decision and that decision makers will avoid choosing alternatives that hurt them politically. Using systemist theory, I compare two journal articles that use poliheuristic theory to explain foreign policy behavior and choices. More specifically, I compare Özdamar and Erciyas's 2020 *Foreign Policy Analysis* article, which uses case study methods to analyze Turkish decisions during the crises of 1964, 1967, and 1974, with Redd's 2002 *Journal of Conflict Resolution* article that uses experimental methods to analyze decision making in an advisory group setting. Systemism uses diagrams in a visual approach to explicate the relationships among various factors in any given theory. As such, systemism enables us to precisely examine how poliheuristic theory has evolved over nearly twenty years as well as compare what the different methodologies of case studies and experimental methods have to offer in explaining the foreign policy behavior of leaders and their advisers.

**Keywords:** noncompensatory decision strategies; poliheuristic theory; political and domestic politics primacy; Turkish foreign policy; two-stage information processing



## 1. Introduction

For nearly thirty years, the poliheuristic theory of decision making has been used by scholars to explain leaders' decision-making processes and choices. Poliheuristic theory has been applied to fields of study such as terrorism (see, e.g., Mintz et al. 2006; Chatagnier et al. 2012), international bargaining (see, e.g., Astorino-Courtois and Trusty 2000; Beckerman-Boys 2014), the diversionary use of force (see, e.g., Mintz 1993; DeRouen 2001), coalition formation (see, e.g., Mintz 1995), international environmental agreements (see, e.g., Below 2008, 2009), nuclear proliferation (see, e.g., O'Reilly 2012), how foreign policy decisions are made at the domestic level (see, e.g., Brummer 2013; Redd 2002; Mintz 2004; Kinne 2005; Goertz 2004; Mintz 2005; Dacey and Carlson 2004; Keller and Yang 2008, 2016; Christensen and Redd 2004; Mintz et al. 1997), and international crises (see, e.g., Mintz 1993; Redd 2005; DeRouen 2003; DeRouen and Sprecher 2004; Kinne 2005; Keller and Yang 2009; Sandal et al. 2011; Taylor-Robinson and Redd 2003; James and Zhang 2005; Özdamar and Erciyas 2020; Westcott 2019; Ye 2007), as well as many others. The studies cited above have also used different methodological approaches to test the tenets of poliheuristic theory, including experimental, formal, statistical (large-N), and case study methods.

I apply the systemist graphic approach to compare Özdamar and Erciyas's (2020) *Foreign Policy Analysis* article, which uses case study methods to analyze Turkish decisions

during the crises of 1964, 1967, and 1974, with Redd's (2002) *Journal of Conflict Resolution* article that uses experimental methods to analyze decision making in an advisory group setting. I aim to show the different applications of poliheuristic theory to important questions in International Relations and foreign policy analysis, both from conceptual as well as methodological perspectives. A systemist theory lens allows us to compare and contrast the application of poliheuristic theory to the Turkish crises of 1964, 1967, and 1974 (Özdamar and Erciyas 2020) with Redd's (2002) use of poliheuristic theory to explain the influence of advisers on foreign policy decision making and choice.

Systemism uses diagrams in a visual approach to explicate the relationships among various factors in any given theory. James (2023, p. 32) states that "Systemism advocates (a) designation of boundaries for a social system, which in turn identify its surrounding environment; (b) statement of macro (i.e., aggregate) and micro (i.e., actor) levels within the system; and (c) complete specification of all possible types of theoretical linkages for a given social system. Item (c) includes the following connections: (i) macro-macro; (ii) micro-micro; (iii) macro-micro; (iv) micro-macro; (v) environment into system; and (vi) system into environment".

Text in each figure is typed in UPPER- or lower-case characters. UPPER-case characters are used for MACRO-level variables while lower-case characters are used for micro-level variables. Each figure comes in double frames—the outer one refers to the environment, and the inner one refers to the system. A system is simply a "comprehensive set of relationships. Units, alone or in combination with each other, create a set of causes and effects that determine how a system operates" (James 2023, p. 7). James (2023, p. 12) provides an easy-to-understand example of the relationship between a system and its environment, as well as macro and micro-level relationships: "Imagine Europe as the system. The macro level would correspond to actions back and forth between and among the governments, along with transactional actors such as the European Union. The micro level pertains to what occurs within the units—namely, individual states and their societies. Outside of the region is the environment—the greater global system. The environment can be expected to provide inputs into and experience outputs from, for instance, the European system".

For both studies included in this article, the environment is the World Beyond. In contrast, for Redd (2002), the system is International Relations, whereas, for Özdamar and Erciyas (2020), it is Political Science. The system for Redd (2002) is related to International Relations primarily because his experimental tests of poliheuristic theory apply to world leaders and how we can thus expect them to process information en route to choices pertaining to foreign policy. The system for Özdamar and Erciyas (2020) is associated with the Political Science system because their application of poliheuristic theory is at the level of Turkish leaders and their administrations across three different discrete crises.

James (2023, p. 15) further notes that "Color and shape are used to designate roles for variables. An initial variable takes the form of a green oval, whereas a terminal variable is depicted as a red octagon. [A] generic variable appears as a plain rectangle. A blue parallelogram designates a point of convergence, and an orange diamond denotes a point of divergence, for pathways. A purple hexagon denotes both convergence and divergence—a nodal variable." Line segments of various types connect these variables, of which the two relevant ones for this paper are solid and dashed. A solid line denotes a connection between two variables explicitly made by the author, whereas a dashed line signifies a connection inferred by the reader but not made explicit by the author. These lines represent linkages, or connections, between two variables (see James 2019, 2023). James (2019, p. 782, fn 6) further states that "Bunge (1996) does not provide instructions on how linkages might be 'unpacked' in the course of explaining each in turn. The specific orderings and combinations of linkages that appear in each set of figures reflect a pragmatic effort toward ease of explanation". A fuller explanation of systemist graphics and notation can be found in the introductory article in this Special Issue, as well as in James (2019, 2023) and Gansen and James (2021, 2023).

As James (2019, p. 781) notes, "the overall value of systemism is that its visual representations clarify relationships expressed in a theory and thereby facilitate constructive criticism and potential scientific advancement." James (2023, pp. 28–29) makes a detailed case for the use of systemism and its accompanying graphic illustration by spelling out its benefits: "First, the graphic is able to probe for logical consistency in a way that is much more direct than through the use of words alone. Second, the graphic conveys a relatively complete treatment in terms of levels of analysis. Third, [there is] the potential for elaboration of the framework once its limitations, as well as contributions, are identified. Fourth, consider the fact that an alternative version of the framework might be created— something that seems by intuition like a negative rather than positive feature. In fact, this is a *strength* of the systemist technique (emphasis in original). Fifth, and finally, diagram[s] can be compared readily to other systemist graphics".

I, along with assistance from the editors of this Special Issue, Sarah Gansen and Patrick James, created a systemist graphic for the Redd (2002) and Özdamar and Erciyas (2020) articles in order to explicate the specific theoretical arguments that each made and the findings they arrived at using the poliheuristic theory of decision making. Using the poliheuristic theory of foreign policy decision making and process-tracing techniques, in an experimental setting, Redd (2002) examines the effects of the presence of advisers on strategy selection and the influence of strategy selection on choice in a hypothetical foreign policy scenario. His findings show that decision makers are highly sensitive to—and cognizant of—the political ramifications of their decisions. Likewise, Özdamar and Erciyas (2020) use poliheuristic theory; however, they focus more specifically on the two-stage process and noncompensatory principle to explain Turkish decision making and choices during the 1964, 1967, and 1974 crises with Greece over control of Cyprus. Their findings show that domestic political factors influenced decision making and choices in a two-stage process in all three crises. Moreover, they also found that factors outside of the direct confines of poliheuristic theory influenced leaders' decision making and choices.

I first provide a brief overview of poliheuristic theory, after which I use systemist theory to analyze each article separately. After reviewing the two articles, I then use systemist graphics to help illustrate points of convergence and divergence in the two applications of poliheuristic theory in these two articles. I then conclude with an evaluation of the benefits of applying systemist theory to our understanding of foreign policy decision making using poliheuristic theory.

## 2. Poliheuristic Theory

Basically speaking, the poliheuristic theory posits that individuals resort to cognitive shortcuts when processing information. The theory makes two assertions. First, individuals use a mixture of decision strategies en route to a *single* choice. Specifically, decision makers employ a two-stage decision process where, in the first stage, they resort to simplifying heuristics, or cognitive shortcuts, in an attempt to alleviate cognitive strain emanating from personal, environmental, and situational factors. In the second stage, decision makers tend to employ more rational and expected utility-maximizing strategies or lexicographic decision rules. The second assertion posits that decision makers use different heuristics (i.e., decision strategies) in response to *different* decisions as a function of environmental and personal variations. In addition to positing that individuals use a two-stage process and multiple heuristics in foreign policy decision making, the poliheuristic theory emphasizes the political aspects of decision making in foreign policy contexts. The assumption is that decision makers measure costs and benefits, risks and rewards, gains and losses, and success and failure in terms of political considerations above all else (Mintz et al. 1997).

Relatedly, poliheuristic theory posits that decision makers utilize five main information processing characteristics en route to making a choice: (1) nonholistic search, (2) dimension-based processing, (3) noncompensatory decision rules, (4) satisficing behavior, and (5) order-sensitive search (Mintz et al. 1997).[1] These information processing characteristics can be thought of as the heuristics that decision makers resort to in order to deal with

environmental/situational and personal/cognitive constraints. These five information processing characteristics will likewise influence the final choices that decision makers arrive at.

### 3. Testing Poliheuristic Theory Propositions Experimentally

Redd (2002) expands on the basic tenets of poliheuristic theory by incorporating the influence of advisers on foreign policy decision making and thus making the theory more relevant to real-world foreign policy decisions wherein advisers provide information and assessments of alternatives along, for example, political, diplomatic, economic, and military dimensions. Redd (2002) then tests for the influence of advisers on foreign policy processes, specifically for dimension- vs. alternative-based processing and noncompensatory decision strategies, as well as for the effects of strategy selection on choice, using process-tracing techniques in an experimental setting. Redd (2002) also tests for order effects, which is one of the five processing characteristics of poliheuristic theory, wherein advisers appear in different sequences in the decision matrix. Figure 1 illustrates the specific arguments and findings from Redd (2002) using systemist notation and graphics.

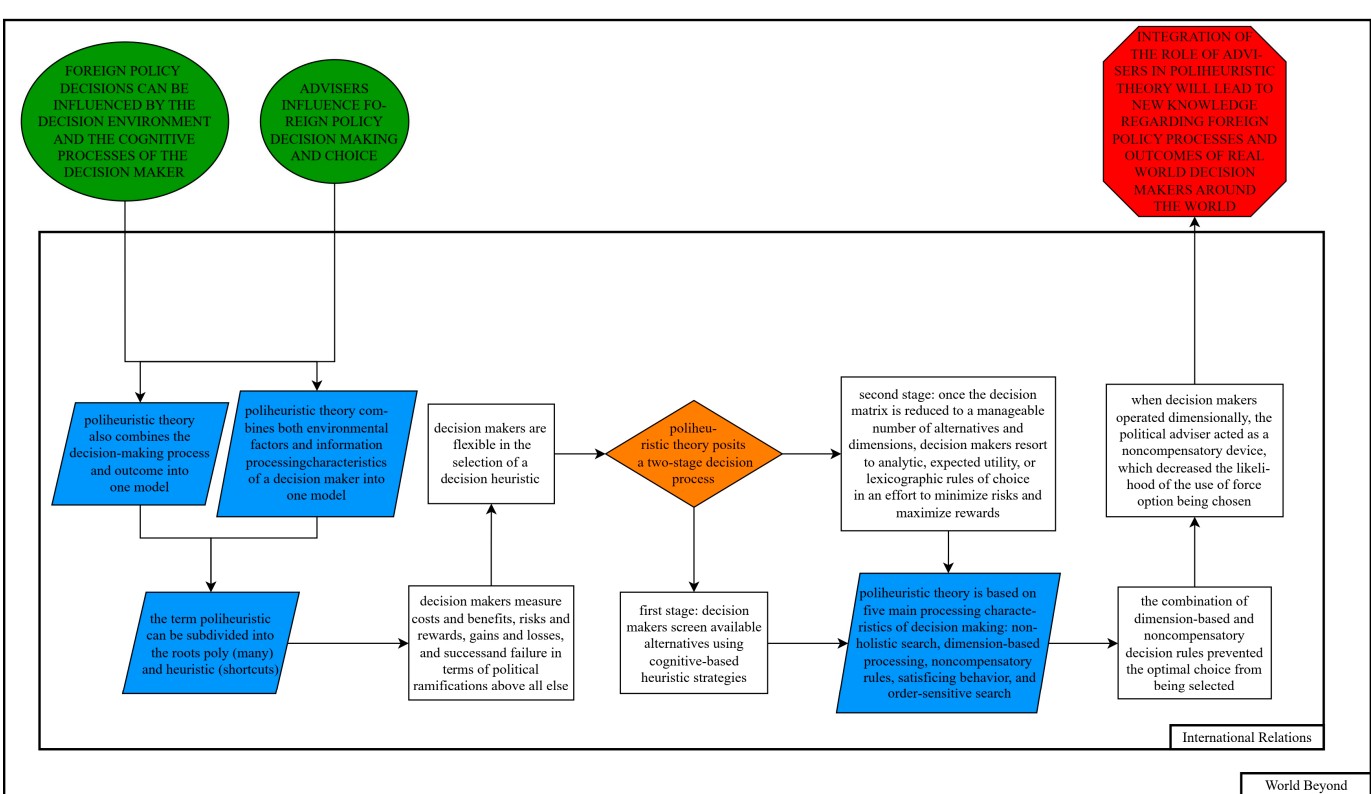

**Figure 1.** The Influence of Advisers on Foreign Policy Decision Making: An Experimental Study (Steven B. Redd 2002) Diagrammed by Steven B. Redd, Sarah Gansen, and Patrick James.

Redd (2002) begins with two initial variables (the green ovals) that represent the beginning point for both poliheuristic theory in general (FOREIGN POLICY DECISIONS CAN BE INFLUENCED BY THE DECISION ENVIRONMENT AND THE COGNITIVE PROCESSES OF THE DECISION MAKER), as well as Redd's more specific and novel contribution to poliheuristic theory, which posits that ADVISERS INFLUENCE FOREIGN POLICY DECISION MAKING AND CHOICE.[2]

With the two initial variables in place, systemist graphics can be used to identify two fundamental theoretical assumptions of poliheuristic theory ("poliheuristic theory also combines the decision-making process and outcome into one model") and ("poliheuristic theory combines both environmental factors and information processing characteristics

of a decision maker into one model"), represented as convergent variables (blue parallelograms). These two convergent variables then flow into a third convergent variable ("the term poliheuristic theory can be subdivided into the roots poly (many) and heuristics (shortcuts)"), which is another key theoretical assumption in poliheuristic theory. These three convergent variables all flow from the broader literature on decision making that examines the interaction of environmental and person-specific factors and the important role of advisers in affecting information processing and choice.

The three abovementioned convergent variables then lead to two successive generic variables that address unique aspects of poliheuristic theory, including the importance of the political dimension in information processing and that decision makers are flexible in the adoption of decision heuristics when en route to a choice. The next step is a divergent variable (orange diamond), which represents one of the most important general hypotheses in poliheuristic theory, wherein decision makers engage in a two-stage decision process. The first stage is characterized by a process wherein decision makers, when faced with either environmental or personal cognitive constraints, resort to heuristic strategies in an attempt to simplify the decision task. In the second stage, once the decision task is reduced to a manageable number of alternatives and dimensions, decision makers resort to either analytic and expected utility or lexicographic rules of choice in an effort to minimize risks and maximize rewards.

These two generic variables, representing the first and second stages of the two-stage process of decision making in poliheuristic theory, then converge into a variable representing the five main processing characteristics of poliheuristic theory as noted above (blue parallelogram). From here, two additional hypotheses that are more specific to Redd's (2002) study are specified, including the assertion that particular characteristics of the political adviser would make it more likely that decision makers would use dimension-based and noncompensatory strategies en route to making a choice and that the order in which information from the political adviser is accessed would affect information processing and choice.

The systemist diagram, in the World Beyond, concludes with a red octagon, symbolizing that the integration of the role of advisers in poliheuristic theory will lead to new knowledge regarding foreign policy processes and outcomes. More specifically, Redd (2002, p. 356) "found that decision makers are highly sensitive to and cognizant of the political ramifications of their decisions. Specifically, the quasi-experimental aspects of the study showed that the political adviser acted as a sensitizing mechanism for decision makers, which led them to adopt noncompensatory decision rules". Additionally, Redd (2002, p. 356) found an interaction effect between the importance of advisers and employed decision strategies, noting that "when decision makers used alternative-based strategies, the distribution of the importance of advisers was inconsequential in aiding decision makers in arriving at the accurate choice. However, when decision makers employed dimension-based and noncompensatory strategies, the importance of advisers became statistically significant. [In other words], the political dimension acts as a noncompensatory sensitizing device for decision makers en route to choice".

## 4. Application of Poliheuristic Theory to Real-World Foreign Policy Crises

Özdamar and Erciyas (2020) use poliheuristic theory to explain Turkey's decision making during the Cyprus crises of 1964, 1967, and 1974. They focus on the two-stage process of poliheuristic theory, arguing that, during each crisis, "Turkish leaders avoided political loss by using the noncompensatory rule" in the first stage, and that, in the second stage, "Turkish leaders wanted to maximize benefits and make the ultimate choice among the remaining options (Özdamar and Erciyas 2020, p. 461). Using primary sources and a case study approach, they found support for both hypotheses across all three crises. Figure 2 illustrates the specific arguments and findings from Özdamar and Erciyas (2020) using systemist notation and graphics.

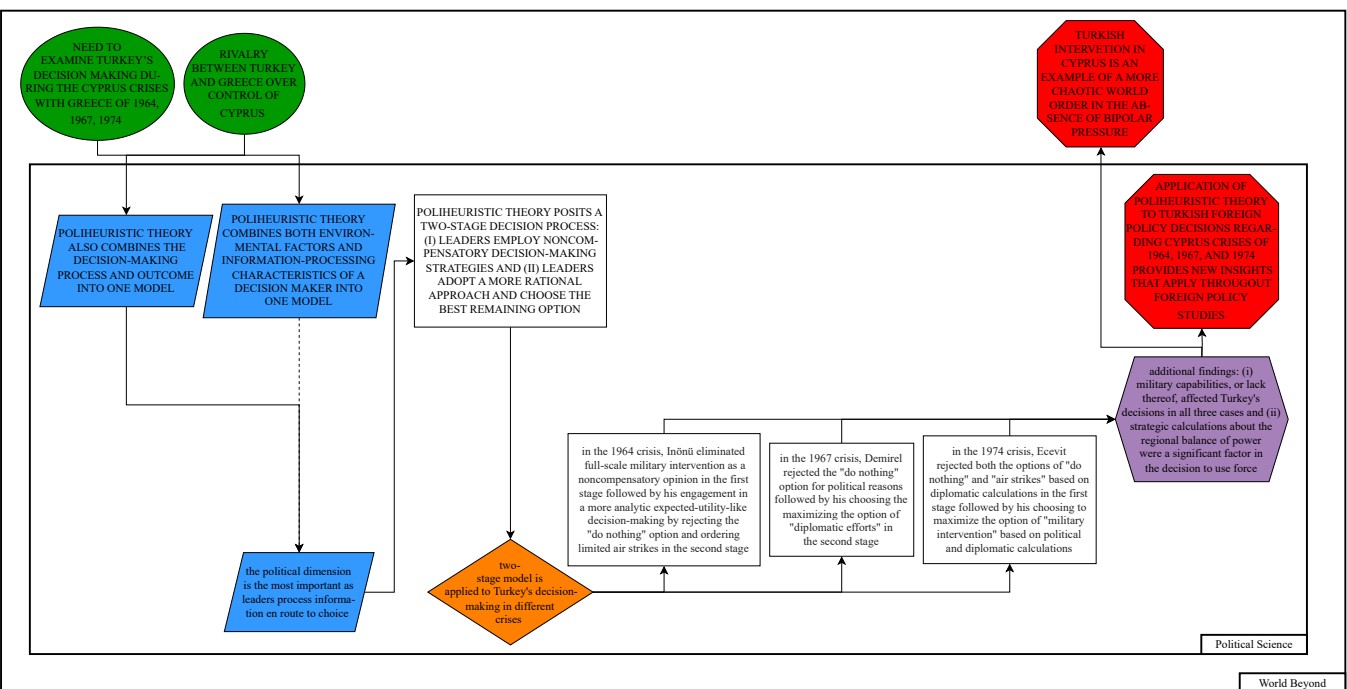

**Figure 2.** Turkey and Cyprus: A Poliheuristic Analysis of Decisions during the Crises of 1964, 1967, and 1974 (Özdamar and Erciyas 2020). Diagrammed by Steven B. Redd, Sarah Gansen, and Patrick James.

Özdamar and Erciyas (2020) begin with two initial variables (the green ovals) that represent the beginning points for their study (RIVALRY BETWEEN TURKEY AND GREECE OVER CONTROL OF CYPRUS and NEED TO EXAMINE TURKEY'S DECISION MAKING DURING THE CYPRUS CRISES WITH GREECE OF 1964, 1967, AND 1974). With these two initial variables in place, systemist graphics help us to identify two applications of poliheuristic theory in Özdamar and Erciyas's (2020) study (POLIHEURISTIC THEORY ALSO COMBINES THE DECISION-MAKING PROCESS AND OUTCOME INTO ONE MODEL and POLIHEURISTIC THEORY COMBINES BOTH ENVIRONMENTAL FACTORS AND INFORMATION PROCESSING CHARACTERISTICS OF A DECISION MAKER INTO ONE MODEL), which are represented as convergent variables (blue parallelograms).[3]

These two variables then lead to another variable that Özdamar and Erciyas (2020) emphasize in their paper ("the political dimension is the most important as leaders process information en route to choice"), which is represented as a blue parallelogram. This variable then means that POLIHEURISTIC THEORY POSITS A TWO-STAGE DECISION PROCESS: (1) LEADERS EMPLOY NONCOMPENSATORY DECISION-MAKING STRATEGIES AND (2) LEADERS ADOPT A MORE RATIONAL APPROACH AND CHOOSE THE BEST REMAINING OPTION, which comprises Özdamar and Erciyas's (2020) two main hypotheses. This variable then leads to the divergent variable (orange diamond), wherein the "two-stage model is applied to Turkey's decision making in different crises".

This divergent variable (orange diamond) then leads to the three main findings that Özdamar and Erciyas (2020) identify for each of the three Cyprus crises and that support poliheuristic theory, and, more specifically, the two-stage process and noncompensatory rule. In the 1964 crisis, "İnönü eliminated the full-scale military intervention as a non-compensatory option because of the immense foreign policy problems it would create for Turkey. In the second stage ... İnönü engaged in more analytical expected-utility-like decision-making" (Özdamar and Erciyas 2020, p. 465). In this second stage, Özdamar and Erciyas (2020, p. 465) note that "'do nothing'" was not a feasible option because it would cause a stronger domestic opposition to the government in the parliament. When all diplomatic efforts failed, İnönü ordered limited air strikes." In the 1967 crisis, Demirel

rejected the "do nothing" option in the first stage for political reasons, followed by his choosing of the maximizing option of "diplomatic efforts" in the second stage. Finally, in the 1974 crisis, Ecevit rejected both the options of "do nothing," and "air strikes" based on diplomatic calculations in the first stage, followed by his choice to maximize the option of "military intervention" in the second stage based on political and diplomatic calculations.

From here, the diagram takes an interesting step into a nodal variable (purple hexagon), which denotes both convergence and divergence. Özdamar and Erciyas's (2020) analysis of the three crises leads to additional findings (results that are somewhat out of the direct purview of poliheuristic theory but which are, nonetheless, quite important): (i) military capabilities, or lack thereof, affected Turkey's decisions in all three cases, and (ii) strategic calculations about the regional balance of power were a significant factor in the decisions to use force. Özdamar and Erciyas (2020, p. 471) "argue that this rather bold decision of the Ecevit administration [to militarily intervene in the 1974 crisis] was made possible by two major factors: the easing of bipolar pressures on Turkey's choices and improvements in military capabilities".

Again, rather interestingly, the systemist diagram concludes with two red octagons, one pitched at the level of "Political Science" as a discipline and one in the "World Beyond." At the "Political Science" level, we find that the APPLICATION OF POLIHEURISTIC THEORY TO TURKISH FOREIGN POLICY DECISIONS REGARDING THE CYPRUS CRISES OF 1964, 1967, AND 1974 PROVIDES NEW INSIGHTS THAT APPLY THROUGHOUT FOREIGN POLICY STUDIES. Özdamar and Erciyas (2020) found that "Turkey's decisions in 1964, 1967, and 1974 clearly demonstrate that the Turkish leadership eliminated all those alternatives deemed 'unacceptable' from the domestic political perspective. In the second stage, utility maximizing, cost minimizing strategic calculations followed. The Turkish case shows that the domestic political dimension—political survival concerns—has been the most important one in the first stage, as suggested by the larger literature (Mintz 1993, 2002; DeRouen 2003; Geva and Mintz 1997; Astorino-Courtois and Trusty 2000; Sathasivam 2003)".

The second red octagon, pitched in the "World Beyond", denotes that the TURKISH INTERVENTION IN CYPRUS IS AN EXAMPLE OF A MORE CHAOTIC WORLD ORDER IN THE ABSENCE OF BIPOLAR PRESSURE. Referring specifically to the 1974 decision to militarily intervene in the Cyprus crisis, Özdamar and Erciyas (2020, p. 473) note that, "After détente began in 1969, Turkey felt less US pressure as well as a lower level of Soviet threat," and that "Strategic calculations about the regional balance of power were a significant factor in the decision to use force".

## 5. Systemist Comparison

In this last section, prior to the conclusion, I highlight some of the more important theoretical and methodological insights provided using systemist analysis and graphics. Systemist notation and graphics are particularly helpful in our efforts to compare and contrast the manner in which poliheuristic theory was used by the authors to explain leaders' foreign policy decision making in the experimental study by Redd (2002) and the real-world case study by Özdamar and Erciyas (2020) of the Cyprus crises of 1964, 1967, and 1974. The technique applied is systematic synthesis.

To begin, the initial starting points for each article as denoted by systemist graphics help clarify the primary purpose of each piece. Redd's (2002) *JCR* article attempts to test some of the primary assertions of poliheuristic theory (FOREIGN POLICY DECISIONS CAN BE INFLUENCED BY THE DECISION ENVIRONMENT AND THE COGNITIVE PROCESSES OF THE DECISION MAKER) as well as incorporate literature on the importance of advisers in studying foreign policy analysis into poliheuristic theory itself (ADVISERS INFLUENCE FOREIGN POLICY DECISION MAKING AND CHOICE). In contrast, Özdamar and Erciyas (2020) use poliheuristic theory to analyze the RIVALRY BETWEEN TURKEY AND GREECE OVER CONTROL OF CYPRUS and, more specifically, to explain TURKEY'S DECISION MAKING DURING THE CYPRUS CRISES WITH GREECE

OF 1964, 1967, 1974. In the latter study, the theoretical propositions of poliheuristic theory, particularly those related to the two-stage process and the use of noncompensatory strategies, are taken to be true and then used to analyze real-world Turkish decision making; however, in Redd's article (2002), the author is trying to test these propositions experimentally. Moreover, while Redd (2002) is testing the effects of advisers in foreign policy decision making and on choice, Özdamar and Erciyas (2020) focus primarily on the three Turkish prime ministers in power during each of the three crises and do not elaborate on any specific influence any advisers may have had on the deliberations regarding foreign policy.

Both studies then proceed to lay out two of the basic theoretical assumptions of poliheuristic theory (the two blue parallelograms directly beneath the two green ovals); however, there are some important differences to note. In Redd's (2002) study, these assumptions are pitched at the micro level, whereas, in Özdamar and Erciyas (2020), they are at the macro level. This difference is a manifestation of the Redd (2002) study being a test of poliheuristic theory, whereas Özdamar and Erciyas (2020) are applying poliheuristic theory to their study of Turkish decision making in the real world. In other words, the assumptions in one study are taken from the discipline as a whole (i.e., macro), while, in the other, they are specific to the study itself (i.e., micro).

The next step in the diagram is different for both. Again, Redd (2002) specifies further assumptions of poliheuristic theory (the term poliheuristic can be subdivided into the roots poly (many) and heuristic (shortcuts)—blue parallelogram; decision makers measure costs and benefits, risks and rewards, gains and losses, and success and failure in terms of political ramifications above all else—white rectangle; and decision makers are flexible in the selection of a decision heuristic—white rectangle). Özdamar and Erciyas (2020) are more specific about which assumption is most important to their study in the following statement: "the political dimension is the most important as leaders process information en route to choice" (blue parallelogram). Another key difference in this part of the diagram is the link between the assumption that "poliheuristic theory combines both environmental factors and information processing characteristics of a decision maker into one model", whether at the micro or macrolevel, to the next step in the respective diagrams. Redd (2002) clearly discusses this assumption, whereas Özdamar and Erciyas (2020), who focus more pointedly on the two-stage aspects of the model and noncompensatory strategies, do not; hence, the dashed line in the Özdamar and Erciyas (2020) piece, an inference made by this author, contrasts with the solid line in Redd's (2002) article.

In Redd's (2002) article, the next major step in the diagram is a central proposition of poliheuristic theory, namely, the assertion that decision makers engage in a two-stage decision process, as noted by the divergent variable in the systemist diagram (orange diamond): in the first stage, they screen available alternatives using cognitive-based heuristic strategies, often referred to as cognitive "shortcuts." In the second stage, once the decision matrix is reduced to a manageable number of alternatives and dimensions, decision makers resort to either analytic and expected utility or lexicographic rules of choice in an effort to minimize risks and maximize rewards. In contrast, in Özdamar and Erciyas' (2020) paper, this same step in the process is pitched at the macrolevel (white box) and is specific, wherein they assert that the POLIHEURISTIC THEORY POSITS A TWO-STAGE DECISION PROCESS: (I) LEADERS EMPLOY NONCOMPENSATORY DECISION-MAKING STRATEGIES AND (II) LEADERS ADOPT A MORE RATIONAL APPROACH AND CHOOSE THE BEST REMAINING OPTION.

Redd (2002) next lists the five main processing characteristics of poliheuristic theory (blue parallelogram), which Özdamar and Erciyas (2020) exclude from their article, with the exception of highlighting the noncompensatory principle in a much earlier step in their systemist diagram. This difference, once again, can be attributed to the fact that the major thrust of Redd's (2002) article is its exclusive focus on poliheuristic theory and the experimental testing of some of its propositions, whereas Özdamar and Erciyas (2020) seek to apply poliheuristic theory to Turkish decision making in the context of the Cyprus crises. In the former (Redd 2002), a more detailed explication of poliheuristic theory is warranted,

while Özdamar and Erciyas (2020) focus, in their study, on the more important aspects of poliheuristic theory and their application in a real-world setting.

I next turn to the findings of the two studies. Redd (2002) finds that decision makers are (1) more likely to use dimension-based, and/or noncompensatory strategies based on the characteristics of the information received from the political adviser, and (2) that the order in which information from the political adviser is accessed affects information processing and choice. Notice that three of the five processing characteristics (dimension-based processing, noncompensatory strategies, and order-sensitive search) as discussed by Redd (2002) are confirmed in the experimental study, along with the importance of the political dimension. The next step in the systemist diagram of Özdamar and Erciyas (2020) is the application of the two-stage model to Turkey's decision making in the three separate Cyprus-related crises (orange diamond), after which a brief synopsis of poliheuristic decision making is provided for each Turkish prime minister and his administration for the 1964, 1967, and 1974 crises, respectively, as discussed earlier. The collective findings from the three crises illustrate that, in the first stage, noncompensatory calculations ruled out certain options primarily based on political and domestic factors; moreover, in the second stage, the Turkish decision makers engaged in maximizing behavior, which also focused on political and diplomatic considerations.

Two particularly interesting findings in the Özdamar and Erciyas (2020) article are represented in the purple hexagon, a nodal variable, wherein (i) military capabilities, or lack thereof, affected Turkey's decisions in all three cases, and where (ii) strategic calculations about the regional balance of power were a significant factor in the decision to use force. These two findings are interesting because neither was hypothesized by Özdamar and Erciyas (2020); yet, they are quite important in the overall explanation of Turkish decision making in all three crises. That is not to say that these two considerations, Turkish military capabilities and strategic calculations about the regional balance of power, or, more directly, the role of the United States in the region, could not have been incorporated into a poliheuristic model. However, Özdamar and Erciyas (2020) did not do so, instead focusing more on the importance of two-stage processing and the noncompensatory principle, and only reporting on the importance of military capabilities and strategic calculations in their analysis of each case and in their conclusion. Without the precise analysis offered by using systemist graphics and notation, this important point could easily be missed.

Redd (2002) ends with a single terminal variable (red octagon) found in the World Beyond denoting that the INTEGRATION OF THE ROLE OF ADVISERS IN POLIHEURISTIC THEORY WILL LEAD TO NEW KNOWLEDGE REGARDING FOREIGN POLICY PROCESSES AND OUTCOMES OF REAL WORLD DECISION MAKERS AROUND THE WORLD.

This ending is a recognition that the primary purpose of the Redd (2002) article was to expand poliheuristic theory by incorporating the literature on the importance of advisers into the model and then testing it experimentally for process and outcome validity.

In contrast, the Özdamar and Erciyas (2020) article ends with two terminal variables (red octagons), with one in Political Science and the other in the World Beyond. In the former, the APPLICATION OF POLIHEURISTIC THEORY TO TURKISH FOREIGN POLICY DECISIONS REGARDING THE CYPRUS CRISES OF 1964, 1967, AND 1974 PROVIDES NEW INSIGHTS THAT APPLY THROUGHOUT FOREIGN POLICY STUDIES. The primary purpose of Özdamar and Erciyas' (2020) article was to explain the how and why of Turkish decision making during these three crises, and they believed that poliheuristic theory would provide these new insights, not only for these three crises, but perhaps also for the study of other decision making in times of crisis. Their second terminal variable in the World Beyond states that TURKISH INTERVENTION IN CYPRUS IS AN EXAMPLE OF A MORE CHAOTIC WORLD ORDER IN THE ABSENCE OF BIPOLAR PRESSURE. This statement alludes to the broader finding that Özdamar and Erciyas (2020) discuss in their paper about the role Turkey and other middle powers played on the world stage in the context of the bipolar nature of the Cold War. More specifically, in the 1974 crisis, when

Turkey intervened militarily in the Cyprus crisis, it did so because the tight bipolarity that had existed previously began to loosen and the United States was no longer pressuring Turkey, as well as because there was a lessened Soviet threat after détente began in 1969 (Özdamar and Erciyas 2020, pp. 472–73). Perhaps there are other real-world examples of middle powers having a greater license to flex their military muscles at this point in history because of this more chaotic world order.

The utility and benefit of using a systemist approach can be summarized by noting that the graphic representation of both articles helps to, at least, achieve three things: (1) we can clearly see how the variables in each article are connected to each other; (2) we also know more precisely how each variable is conceptualized in each article; (3) we can therefore more readily compare the similarities and differences between the two different applications of poliheuristic theory and do so with a level of precision not afforded through the use of words alone. In the comparison of the Redd (2002) and Özdamar and Erciyas (2020) articles, graphic representation helps us see that poliheuristic theory was modeled differently, wherein Redd (2002) includes some concepts/variables in the model that Özdamar and Erciyas (2020) omit and vice versa. Finally, we can also quickly see the flow of the theoretical argument in each paper, as well as all of the connections between variables in both the micro and macro levels and the relationships between these levels.

## 6. Conclusions

Systemist graphics are particularly beneficial in helping researchers uncover the particular ways in which different scholars explicate specific theoretical constructs, methodological applications, and unique findings. In this study, I used systemist graphics (James 2019; Gansen and James 2021) to compare two different studies (Özdamar and Erciyas 2020; Redd 2002), using poliheuristic theory to explain foreign policy behavior. The results provide a clearer picture of the different ways in which poliheuristic theory has been used to explain foreign policy decision making, with specific emphasis on the two-stage processing, noncompensatory principle, order-sensitivity, and political primacy aspects of the theory. Furthermore, systemist graphics also helped to highlight specific findings from the theory, as well as to differentiate those findings from broader themes in the literature on the influence of those factors outside of the decision maker and the decision environment, such as the world in which middle powers operate. Finally, we gain a better understanding of the interaction of the theoretical aspects of the theory from the methods used to test those theoretical propositions. Systemist graphics have aided us in better understanding the link between the experimental tests of poliheuristic theory versus the application of poliheuristic theory to the real-world cases of three Turkish foreign policy crises.

**Funding:** This research received no external funding.

**Institutional Review Board Statement:** Not applicable.

**Informed Consent Statement:** Not applicable.

**Data Availability Statement:** Data sharing not applicable.

**Conflicts of Interest:** The author declares no conflict of interest.

## Notes

[1]  See Redd (2000) for a more thorough explanation of each of these five decision-making characteristics.

[2]  While the phrase "initial variables" is used here and in the discussion of the Özdamar and Erciyas (2020) piece, they are not always "variables" in the strict sense of the word, wherein the attributes or characteristics of a concept can vary across observations (see Forestiere 2022). Rather, as James (2019, p. 784) notes, an initial variable is "The starting point in a series of relationships." James (2023, p. 11, see fn 1) further states: "The language used to introduce the basic systemist [figure] refers to variables, which is conventional in the context of the quest for explanation. Other scholars, more oriented toward normative analysis, will refer to components of a systemist figure as stages in an argument rather than variables". The author thanks one of the anonymous reviewers for pointing out this important, and clarifying, distinction.

³    I will explain in the comparison section of the paper why these two variables—and others—are capitalized here in the Özdamar and Erciyas (2020) piece but are in lower case in Redd (2002).

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
