# Peer review of "Key Concepts in the Poliheuristic Theory of Foreign Policy Decision Making: A Comparative Examination Using Systemist Theory"

_socsci, doi:10.3390/socsci12080446_

Round 1

Reviewer 1 Report

This article on poliheuristic theory is so well done I have nothing to add. I was not familiar with the subject but feel I could grasp it clearly from the presentation. This of course is the core goal of the article using the systemist approach.

Author Response

Thank you so much for the positive review!

Author Response

I very much appreciate the three primary suggestions offered by reviewer 2 to improve the manuscript.  I believe the revisions suggested by reviewer 2 have lead to a much better paper, and particularly one that better explains key concepts in systemism and its overall value as an approach to understanding theoretical arguments in general, and more specifically in the context of the comparison of the two articles.

The first point made by reviewer 2 was a request to explain the systemist approach in greater detail.  In response, I have added three full paragraphs found on page 2 wherein I have expounded more fully on the important graphics used in systemist theory and its accompanying diagrams.  This includes additional explanations regarding the text boxes and their colors and what they represent, the line segments that connect variables, and the relationships among variables at the micro and macro levels.

The second request by reviewer 2 was to more clearly explain the general value of the systemist approach.  I have added a paragraph on pages 2-3 that details more fully the overall value of the systemist approach.

The third and final suggestion made by reviewer 2 was to speak more clearly to the value of systemist theory to the specific application as discussed in the comparison of the two articles.  Reviewer 2 asks, "So what is it that the systemist graph elucidates, specifically?"  And, "What would I not have been able to see (or would it have been difficult to see) in the absence of this framework?"  I have added a paragraph to page 10 that addresses reviewer 2's questions, wherein I enumerate the value added by systemism in the comparison of the two articles.

Again, I thank reviewer 2 for the careful reading of the manuscript and for the suggested changes that will not only serve to improve this paper, but also benefit the special issue of the journal.

Author Response

I am grateful for reviewer 3's thoughtful reading of my manuscript and for the suggested revisions.  I believe that the paper is now much stronger given the suggested changes, and the paper can now not only serve as a standalone paper, but its contributions to the special issue have been improved.  In the paragraphs that follow, I outline in detail the changes I have made as suggested by reviewer 3.

1. Reviewer 3's first concern regards the nature and conceptualization of what are termed "initial variables" in systemist graphics.  I have included an additional footnote, now fn 2 found on page 5, that seeks to clarify the meaning of this term.

2. The second concern regards the frames for the Redd (2002) and Ozdamar and Erciyas (2020) pieces and the distinction between the system and the environment.  I have added additional text found on page 2 that addresses in greater detail what is meant by a "system" and its "environment," and how they are differentiated, as well as related.  An additional concern expressed by reviewer 2 deals with the system in the Redd piece versus the Ozdamar and Erciyas piece and why they are not the same when it would seem that perhaps they should be.  The first point to note is that the designation of a disciplinary system in a systemist diagram is an inductive, case-by-case, type of endeavor.  However, in an attempt to help clarify substantively why they are different, I have added text on page 2, third full paragraph.

3. Reviewer 3 had questions about the nature of the arrows, or line segments.  I have added text on page 2, fourth full paragraph, clarifying the nature and purpose of line segments in systemist graphics.

4. There were concerns about the conclusions for the Redd piece, specifically the last two generic variable boxes, wherein there was a lack of "substantive information or direction for these relationships."  The author especially thanks reviewer 3 for pointing out this problem.  The wording in those text boxes will be changed to actually report findings, which should make it more commensurate with the findings presented in the Ozdamar and Erciyas piece.  Specifically, the boxes will read as follows: "The combination of dimension-based and noncompensatory decision rules prevented the optimal choice from being selected," and "When decision makers operated dimensionally, the political adviser acted as a noncompensatory device, which decreased the likelihood of the use of force option being chosen."

5. Reviewer 3 wonders why the red octagon for the Redd article is placed in the "world beyond" environment rather than in political science/IR academic system.  It should be in the world beyond environment, but in order to more clearly solidify its place there, I have added additional text so that it will now read, "INTEGRATION OF THE ROLE OF ADVISERS IN POLIHEURISTIC THEORY WILL LEAD TO NEW KNOWLEDGE REGARDING FOREIGN POLICY PROCESSES AND OUTCOMES OF REAL WORLD DECISION MAKERS AROUND THE WORLD."

6.a. There should be a dashed line as noted in the text of the paper.  Pat James and Sarah Gansen, editors for the special issue, will rectify this issue since the software that creates the graphics is under their control.

6.b. Again, I thank reviewer 3 for the careful attention to detail in catching this rather clunky prose.  I have broken up the one confusing sentence into three separate sentences, and they can be found on page 7.

Again, I very much appreciate reviewer 3's careful attention to detail, and for the many suggestions that I believe make this a much stronger paper.